# Unexpected Prevalence of *eae*-Positive *Escherichia coli* in the Animas River, Durango, Colorado

**DOI:** 10.3390/ijerph17010195

**Published:** 2019-12-27

**Authors:** Steve Hamner, Steven D. Fenster, Benjamin T. Nance, Katherine A. McLain, Kami S. Parrish-Larson, Michael W. Morrow, Timothy E. Ford

**Affiliations:** 1Department of Environmental Health Sciences, School of Public Health & Health Sciences, University of Massachusetts Amherst, Amherst, MA 01003, USA; 2Department of Microbiology, Montana State University, Bozeman, MT 59717, USA; 3Department of Biology, Fort Lewis College, Durango, CO 81301, USA; sdfenster@fortlewis.edu (S.D.F.); benjamin.nance@rvu.edu (B.T.N.); mt.katmclain@gmail.com (K.A.M.); parrishlarson_k@fortlewis.edu (K.S.P.-L.); 4Department of Biology, University of Montana Western, Dillon, MT 59725, USA; michael.morrow@umwestern.edu

**Keywords:** enterohemorrhagic *E. coli*, enteropathogenic *E. coli*, pathogen detection, waterborne disease

## Abstract

Since 2014, biology students at Fort Lewis College have studied the water quality of the Animas River in Durango, Colorado. Environmental microbiology and molecular biology techniques have been employed to study *Escherichia coli* isolates from the river and to define characteristics of the bacteria related to public health. *E. coli* was found in the river, as well as in culverts and tributary creeks that drain into the river within the Durango city limits. Concentrations of *E. coli* in the river occasionally exceeded the US EPA guideline of 126 CFU per 100 mL for recreational water use. Many of the *E. coli* isolates were able to be grown at 45 °C, an indication of mammalian origin. Unexpectedly, 8% of the isolates contained the intimin (*eae*) gene, a virulence gene characteristic of two pathotypes of *E. coli*, the enterohemorrhagic and enteropathogenic *E. coli.* Several isolates tested were resistant to multiple antibiotics commonly used in animal and human medicine. Further study is warranted to determine the source of these bacteria entering the Animas River, and to further characterize the possible disease potential of multi-antibiotic resistant and virulence gene-containing isolates found in a semi-rural/urban setting.

## 1. Introduction

From its headwaters in the San Juan Mountains in southwest Colorado, the Animas River flows south through Silverton, a historic mining town, and Durango, a city having origins in the mining industry and currently popular as a center for outdoor recreation and tourism. Downstream of Durango, the river flows through the Southern Ute Indian Reservation, and then crosses the state line into northern New Mexico, passing through the communities of Aztec and Farmington before it merges with the San Juan River.

The Animas River is heavily utilized for recreational activities, including swimming, rafting, and fishing. A segment of the river near Durango is designated a “gold medal” fishing destination [1], a label given to streams producing large numbers of trophy trout. Recreational use of the river has a significant economic impact. On the lower Animas River in the Durango area, a 2006 study estimated that whitewater recreation contributed over $19 million to the local economy annually [2].

Poor water quality is a long-standing issue facing the Animas River watershed, with historic mining activity a major contributing factor. High levels of heavy metals leaching into streams both from acid mine drainage and from non-mined natural deposits [3] have resulted in severely impaired benthic invertebrate and fish populations in the upper Animas River [4,5,6]. The legacy of acid mine drainage and the continuing threat to water quality was recently highlighted by the accidental release of an estimated three million gallons of mining wastewater on 5 August, 2015 from the Gold King mine [7]. The toxic wastewater spill turned the entire Animas River a mustard color as the heavy metal-laden spill passed down the entire length of the river, posing a threat to water supplies in three states. Such a spill poses a particular threat to Native American communities that have limited water treatment capacity [8,9,10,11]. In particular, the spill had a major impact on the Navaho tribal community, threatening crops and livestock, and necessitating transportation of water for both irrigation and ranching.

Another significant threat to water quality and public health has been the presence of fecal bacteria of human origin in the Animas River, identified in the lower segment of the river in the vicinity of Aztec and Farmington [12]. In recent years, sporadic failures of infrastructure including sewage pipes [13] and sewage pump stations [14] have released human fecal material into the Animas. In 2016, a sewage lagoon serving a mobile home park and leeching human fecal material into Lightner Creek, a tributary feeding into the Animas River in Durango, was ordered shut down due to long-term violations of the Clean Water Act [15]. The Environmental Protection Agency (EPA) has set guidelines for the amount of *E. coli* that can be present in recreational waters such as the Animas River. In 1986, the EPA established a limit for *E. coli* in recreational waters of 126 CFU per 100 mL of water [16]. This limit is based on the geometric mean of at least five independent samples over a 30 days period. Levels of *E. coli* above 126 CFU per 100 mL have been shown to correlate with the presence of pathogenic microbes and are used as an indicator for public health risk associated with freshwater fecal contamination. Measurements of *E. coli* along the Animas River are of public health interest due to the large number of people using the water recreationally as well as nearby homes.

Most strains of *E. coli* do not cause disease in humans. *E. coli* bacteria are normal inhabitants of the human intestinal tract [17] and are important for vitamin production and in maintaining digestive health. Some *E. coli* can cause illness in humans. Depending on the organ system targeted, different pathotypes of *E. coli* may cause a variety of diseases, including enteric-diarrheal disease, hemolytic-uremic syndrome, neonatal meningitis, septicemia, and urinary tract infections [17]. Since its identification in 1982, serotype O157:H7 has become perhaps the most well-known of the disease-causing *E. coli*, responsible for over 70,000 cases of illness and about 61 deaths in the US annually [18].

The pathotypes tested for in this study are the enterohemorrhagic *E. coli* (EHEC) and enteropathogenic *E. coli* (EPEC). EHEC bacteria, including O157:H7, have the potential to cause life-threatening hemorrhagic diarrhea and hemolytic uremic syndrome (HUS) in humans, while EPEC bacteria are a leading cause of diarrhea in infants [19]. Like other diarrheagenic pathogens, EHEC and EPEC bacteria are transmitted via the fecal-oral route, often by foodborne and waterborne pathways. Both EHEC and EPEC contain the *eae* gene encoding intimin, a protein that promotes bacterial attachment to host intestinal tissue as part of the disease process [20]. Genetic testing of waterborne *E. coli* for the presence of *eae* allows for ready identification of possible EHEC and EPEC strains. While the presence of the intimin gene by itself does not establish virulence in *E. coli*, the gene is nonetheless a key component of both EHEC and EPEC.

Domestic animal production is the major source of fecal contamination worldwide [21]; fecal material from livestock inevitably finds its way into bodies of water through runoff. Cattle are potential disease vectors of particular interest due to the widespread prevalence of ranching operations in the western United States including Colorado. Both EHEC and EPEC can be carried asymptomatically in cattle populations and a variety of wildlife. It has been estimated that about 30% of feedlot cattle harbor the EHEC serotype O157:H7 [22]. Cattle that harbor pathogenic *E. coli* can shed significant amounts of the bacteria in their manure; the rate of shedding is highly variable, depending on the time of year [22]. Another concern is the routine use of antibiotics in animal feed. While certain antibiotics are used to increase weight gain and productivity of cattle husbandry, development of antibiotic resistant strains of pathogenic bacteria poses a risk to human health [23,24]. The regular use of antibiotics in feedlot operations, in combination with runoff and poor waste management, can result in release of antibiotic resistant bacteria into soils and nearby waterways [25,26].

There is no noted history of pathogenic *E. coli* outbreaks related to bacterial contamination of the Animas River, despite the documented presence of fecal bacteria of human origin in the river. Given the reported incidents of human sewage contamination of the river, the present study was initiated with three goals in mind: to begin a targeted study of the river for presence of *E. coli* bacteria that might pose a public health risk; to introduce biology students at Fort Lewis College to participate in an ongoing program of “citizen science” and involvement in environmental monitoring; and to share student research on water quality with local public health officials and community partners.

## 2. Materials and Methods

During the 2014–2015 academic year at Fort Lewis College, water samples were collected for analysis on 30 August, 7 September, 13 October, 30 October, 29 January and 21 February. Three different sites were studied during this period (Figure 1). Geographic coordinates were captured for each sampling site (Table 1) using Google Earth software installed on a smartphone. The first site was the Animas River at Santa Rita Park in Durango. River sampling was conducted about one foot from the bank in gently flowing current, holding the mouth of the sampling container 2–3 inches below the water surface and facing upstream. During sampling, care was also taken to avoid stirring up river or drainage sediment. The second sampling site was a narrow concrete culvert that feeds into the Animas River near Schneider Skate Park (also referred to as the skateboard park). This site is a very shallow drainage (about 3–4 inches water depth). The third sampling site was Junction Creek just upstream of the confluence of Junction Creek and the Animas River. This third site was sampled only on the dates of 30 October, 29 January, and 21 February. All samples were collected in sterile plastic containers, stored on ice, and transported to Fort Lewis College for analysis. Both the Skate Park and Junction Creek sites are upstream of Santa Rita Park.

For each measurement, a 100 mL sample volume was filtered through a 0.45 micron filter. The filter with trapped bacteria was then placed in a sterile petri dish on top of an absorbent pad presoaked with 2 mL m-ColiBlue24 broth (Hach) per the manufacturer’s protocol. m-ColiBlue24 is a differential and selective media, which selects for growth of coliform bacteria but prevents growth of non-coliform bacteria. The media also differentiates between *E. coli* and non-*E. coli* coliform bacteria—*E. coli* colonies appear blue while non-*E. coli* coliforms are red. Dishes were labeled and incubated overnight at 37 °C. All water samples were filtered and plated in triplicate. Dishes were examined the next day by the instructor (Hamner), and then stored at 4 °C until the next lab class period for examination and processing by students.

After incubation, the samples were evaluated visually with the aid of a dissecting microscope to establish a count (CFU per 100 mL sample) of blue-colored (*E. coli*) colonies. Because growth on m-ColiBlue24 identifies presumptive *E. coli* colonies, m-ColiBlue24 can be used as a first step in isolation of *E. coli* from environmental water sources [27]. Using sterile technique, bacteria from blue colonies were transferred using the streak-for-isolation technique to pairs of nutrient agar dishes. One dish from each pair was incubated overnight at 37 °C and the other was incubated at 45 °C as a test for thermotolerance [28,29]. After incubation and growth of colonies, Gram staining was performed to determine if the colonies contained Gram-negative coliform bacteria (indicating *E. coli*). Students were directed to repeat the streak-for-isolation procedure if Gram staining results were not indicative of pure isolates of Gram-negative coliforms.

To prepare DNA for polymerase chain reaction (PCR) testing, cells were picked from presumptive *E. coli* colonies and transferred into 200 μL of sterile distilled water contained in sterile microfuge tubes. The tubes were vortexed to disrupt cells and DNA was released by boiling the tubes for 10 min. The boiled preparations were then centrifuged at 10,000 rpm for 10 min to pellet cell debris. The supernatant fluid consisting of water with dissolved DNA from each preparation was transferred to a fresh, sterile microfuge tube. DNA preparations were tested by PCR for the ß-glucuronidase (*uid*) gene indicative of *E. coli*. The PCR primers used were as described by Ram et al. [30]:

*uid* forward: 5′ -AATAATCAGGAAGTGATGGAGCA-3′

*uid* reverse: 5′ -CGACCAAAGCCAGTAAAGTAGAA-3′

PCR was performed using LA Taq polymerase kit reagents and 6 μL of the DNA-containing supernatants (final reaction volume of 15 μL) for 30 cycles using the following cycling parameters: 60 s at 94 °C, 30 s at 60 °C, and 30 s at 72 °C. PCR products were analyzed by agarose gel electrophoresis to identify isolates generating a 587-nucleotide ß-glucuronidase amplicon.

Bacterial strains were next tested for presence of the intimin (*eae*) gene. The PCR primers used were as described by Chakraborty et al. [31]:

*eae* forward primer: 5′ -AAACAGGTGAAACTGTTGCC-3′

*eae* reverse primer: 5′ -CTCTGCAGATTAACCCTCTGC-3′

PCR was performed as described above, except that the following cycling parameters were used: 60 s at 94 °C, 30 s at 57 °C, and 30 s at 72 °C. PCR products were analyzed by agarose gel electrophoresis for an expected amplicon size of 454-nucleotides for the *eae* gene. Positive- and negative-control DNAs were used and prepared from a known EHEC strain (DEC 3A, strain 3299-85) and a non-pathogenic/non-thermotolerant *E. coli* strain (previously isolated from an environmental source and characterized in the Ford lab), respectively.

During study of bacteria isolated from water samples collected on 21 February 2015, microbiology students in one class also tested thermotolerant isolates of *E. coli* for antibiotic resistance using the disk diffusion method [32]. Overnight cultures of 16 *E. coli* isolates were grown in Luria broth at 37 °C. Confluent bacterial lawns were prepared by inoculating bacteria from the overnight cultures onto nutrient agar plates using sterile applicator swabs. Discs containing antibiotics (Becton Dickinson) were then placed onto each plate, and the plates incubated overnight at 37 °C. Aseptic technique was used during the placement of antibiotic discs, using flame-sterilized forceps, to avoid cross-mixing of antibiotics. The antibiotics tested included ampicillin, ciprofloxacin, penicillin, streptomycin, tetracycline, and vancomycin. Zones of inhibition of growth were evaluated by direct observation.

Follow-up measurements of *E. coli* levels and testing for the presence of *eae*-positive bacteria were conducted in October 2015 by the authors Nance and Hamner. Water samples were collected on 16 October 2015 by Nance for the three sites previously described, as well as for six additional sites as noted in Table 1. *E. coli* concentrations were measured by Nance at Fort Lewis College using m-ColiBlue24 broth as described above. For PCR testing, 50 mL aliquots of water from each site were shipped to the instructor (Hamner) at Montana State University. Samples were filtered and plated using the m-ColiBlue24 protocol, solely for the selective growth of blue *E. coli* colonies for PCR testing. Blue colonies were sampled and regrown on nutrient agar using the streak-for-isolation technique. After overnight incubation at 37 °C, all regrown bacterial isolates were processed and tested for *eae* as described above. Testing for *uid* was not conducted for these isolates. Isolates were additionally tested by PCR for the presence of Shiga toxin genes (*stx1* and *stx2*) as described previously [27]. PCR testing of October 2015 samples was carried out only by the instructor (Hamner) at Montana State University and not by Nance or students at Fort Lewis College.

Individual instructors have conducted follow-up testing of *E. coli* from river sampling as part of several biology course lab sections at Fort Lewis College since the initial lab exercises were introduced in 2014. During the 2018–2019 academic year, coauthors Fenster and Parish-Larson guided students in testing water samples collected during 23–24 October 2018, and 10–12 March 2019. Samples were collected and transported to Fort Lewis College, and processing and testing conducted as described above.

Students in both the lower division introductory biology and the upper division microbiology classes at Fort Lewis College were guided through and completed the exercises of measuring *E. coli* concentrations from Animas River water samples, culturing presumptive *E. coli* bacteria through streaking-for-isolation technique, and PCR testing of isolates for the *uid* and *eae* genes. Only the upper division microbiology students carried out the subsequent exercise to test *E. coli* isolates for antibiotic resistance.

For all work conducted by the students at Fort Lewis College, the instructors closely monitored student work and test results to ensure quality control of the data being generated. Instructors checked and confirmed *E. coli* colony counts, Gram staining results, PCR test results, and antibiotic resistance test results for each student. Student work with techniques such as micropipetting and aliquoting reagents, or in preparing DNA for PCR, could not be monitored in detail by instructors for each step of the procedures with each student, so the quality of work and absence of mistakes could not be completely guaranteed for every student.

In these exercises, because of the potential that bacterial colonies grown up by students might contain the *eae* virulence gene being tested for, and because colonies might express antibiotic resistance, Biosafety Level 2 (BSL-2) practices were followed. Students were introduced to appropriate lab safety procedures, including but not limited to: disinfecting lab bench work surfaces at the start and end of the lab period; employing sterile technique; keeping long hair tied back and away from Bunsen burner flames and Petri dishes containing bacterial growth; wearing gloves, lab coats, long pants, and closed-toe shoes; and use of biohazard/autoclave bags and containers for disposal of contaminated items such as Petri dishes and disposable plastic ware such as pipette tips. In using a positive control strain of *E. coli* containing the *eae* and *stx1*/*stx2* genes, only the instructor (Hamner) handled and grew bacteria, prepared DNA, and performed the PCR for the control reactions.

## 3. Results

During the period August 2014 through February 2015, a total of fifteen samples were collected from the Animas River sites and tested for *E. coli* concentrations (Figure 2). On 29 January and 21 February 2015, *E. coli* concentrations for the Animas River at Santa Rita Park exceeded the EPA guideline of 126 CFU per 100 mL for recreational water use. *E. coli* concentrations at the skateboard park drain site also exceeded 126 CFU per 100 mL on three of the six sampling dates, but the site is of lesser concern in terms of direct human exposure as it is not classified for recreational water use. It remains, however, important as a potential source of contamination to the river. *E. coli* concentrations at the Junction Creek site were below the EPA guideline on each sampling date.

After streaking for isolation and performing Gram staining, most students were able to microscopically observe Gram-negative coliform bacteria, consistent with positive isolation of *E. coli*. Students having mixed cultures of bacteria (mix of Gram-positive and Gram-negative bacteria), or Gram-positive bacteria, and/or non-coliforms were guided in preparing a second round of cultures through repeating the streak-for-isolation procedure as time permitted. Students unable to generate a pure isolate of Gram-negative coliforms were still allowed to carry out PCR testing in order to gain experience with the procedures and the experimental aims. During subsequent testing using PCR protocols, 36 out of 72 students were able to confirm the presence of the *uid* gene for their isolates, supporting the identification of *E. coli* (data not shown). All samples, both *uid*-positive and *uid*-negative, were then tested for the intimin gene. Six isolates of the 36 that were positive for *uid* also tested positive for *eae*. None of the *uid*-negative isolates tested positive for *eae*. All but one of the 72 bacterial isolates grew at 45 °C an indication of thermotolerance and a high likelihood of fecal origin.

In addition to PCR testing, *E. coli* isolates generated from sampling on 21 February 2015 were tested for resistance to six commonly used antibiotics (see Table 2). Of the four isolates obtained from the Santa Rita Park sample, one isolate was resistant to penicillin only (phenotype abbreviated as Pen^R^), two isolates were resistant to both penicillin and vancomycin (Pen^R^Van^R^), and one isolate was resistant to ampicillin, penicillin, and vancomycin (Amp^R^Pen^R^Van^R^). Among the five isolates obtained from the skateboard park drain, four were resistant to ampicillin, penicillin, and vancomycin (Amp^R^Pen^R^Van^R^), while the fifth isolate was resistant to ampicillin, ciprofloxacin, and penicillin (Amp^R^Cip^R^Pen^R^). Among the seven isolates obtained from the Junction Creek site, one was resistant to penicillin and partially resistant to vancomycin (Pen^R^Van^PR^); a second was resistant to only penicillin (Pen^R^); a third was resistant to both penicillin and tetracycline (Pen^R^Tet^R^); a fourth was resistant to penicillin and tetracycline, and partially resistant to vancomycin (Pen^R^Tet^R^Van^PR^); two isolates were resistant to penicillin and vancomycin (Pen^R^Van^R^); and the seventh isolate was sensitive to all six antibiotics. In summary, among the sixteen isolates, one showed no resistance to the six antibiotics tested, two were resistant to a single antibiotic, and thirteen exhibited multi-antibiotic resistance. It is noteworthy that antibiotic resistance profiles (i.e., phenotype patterns) were shared between the Santa Rita Park Animas River isolates and Junction Creek isolates, and between Santa Rita Park Animas River isolates and Skate Park isolates, but not between Junction Creek isolates and Skate Park isolates.

During October 2015, follow-up sampling of additional sites along the Animas River led to identification of additional drainage sites within the Durango city limits that are sources of *E. coli* found in the river (Table 1 and Table 3). Among 77 bacterial isolates obtained from this sampling, five tested positive for the *eae* gene. Additional testing of these five *eae*-positive isolates revealed that none contained the *stx1/stx2* Shiga toxin genes found in EHEC strains. It is thus likely that these *eae*-positive strains are related to the EPEC pathotype lacking Shiga toxin genes.

During October 2018 and March 2019, the three original sites of Junction Creek near its confluence with the Animas River, the Skate Board Park drain, and the Animas River at Santa Rita Park were retested. Given the concern raised in 2016 of the sewage lagoon releasing human sewage into Lightner Creek not far upstream of its confluence with the Animas River, Lightner Creek samples were also collected and tested. During October 2018 testing, 12 of 46 total isolates from all sites were shown to harbor the *eae* gene, while testing in March 2019 revealed that 5 of 45 bacterial isolates contained the *eae* gene (Table 4).

## 4. Discussion

An environmental microbiology study was conducted in Fort Lewis College biology classes during the 2014–2015 school year, to engage students in understanding and appreciating the importance of monitoring the environment to protect public health. Due to widespread recreational use of the Animas River and the abundance of nearby homes and wildlife, the presence of *E. coli* in the river and associated drainages was studied, and *E. coli* isolates tested for virulence-associated traits. Three individual sites were initially sampled along the Animas River in Durango, CO, to allow for measurement of *E. coli* concentrations, isolation of environmental *E. coli* strains, and testing for possible pathogenic strains of *E. coli*. Bacteria isolated from each sample were evaluated for thermotolerance and Gram stain reaction, and tested by PCR for the presence of the *uid* and *eae* genes.

The EPA has established a level of *E. coli* of 126 CFU per 100 mL that is acceptable in what are deemed recreational waters [16,33]. The Santa Rita Park test site on the Animas River sometimes exceeded this level. The drainage near the skateboard park, while it also exceeded the EPA limit, is not classified as recreational water. It is relatively unlikely that people would ingest water from that drain, although a family with children was observed to picnic by and splash in the drain during one sampling visit (Hamner, observation). The Animas River itself is a recreational waterway that is regularly used by tourists and local residents. The goal of the EPA’s established guideline is to help ensure that a recreational waterway does not have a high enough level of fecal contamination that would indicate the presence of pathogenic microbes [33].

Genetic screening of bacteria for the presence of the *uidA* or *gusA* gene encoding ß-glucuronidase (GUR or GUD) is often used for detection of *E. coli* [30,34,35]. About 95% of *E. coli* isolates express GUR activity [30,35]. Accordingly, several methods approved by the EPA for detection and enumeration of waterborne *E. coli*, including the m-ColiBlue24 method used in the present study, are based on GUR activity [36]. In this study, we used m-ColiBlue24 both to measure the concentration of *E. coli* in water samples, and as an initial screening step to study *E. coli* isolates growing on m-ColiBlue24 media. “Most if not all *E. coli* isolates” are believed to “carry sequences for the *uidA* gene regardless of GUD activity” [34]. Some pathogenic *E. coli*, including variants of O157:H7, have mutations in the coding region of the *uidA* gene, producing an inactive GUR [37]. In the past, it was generally thought that all O157:H7 isolates were GUR-negative [30,34]. Use of the m-ColiBlue24 method based on a functional GUR enzyme would thus fail to identify and include GUR-negative strains of O157:H7 during the enumeration of *E. coli*. More recently, GUR-positive strains of O157:H7 have been identified from wildlife [38] and humans [39], indicating that GUR-positive colonies growing on m-ColiBlue24 media may include O157:H7 isolates.

Of the 72 *E. coli* isolates obtained from the three sites during the 2014–2015 school year, seven samples tested positive for the *eae* gene coding for intimin, a virulence component characteristic of two pathogenic forms of *E. coli*, EHEC and EPEC. While this does not prove that the *E. coli* present in the Animas River are indeed pathogenic, testing for the *eae* gene is a means for screening and identifying strains of *E. coli* that may possibly cause human illness. The *eae* gene is a key component of the locus of enterocyte effacement (LEE). Proteins coded for by the LEE pathogenicity island provide for intimate attachment by pathogenic EHEC or EPEC bacteria to the intestinal wall of the host, an early step in the infection process [40]. Verifying the presence of the intimin gene in environmental isolates of *E. coli* indicates the possible presence of EHEC and EPEC.

The discovery of *eae*-positive *E. coli* in the Animas River was an unexpected finding given the lack of cattle or other livestock ranching activity in the areas of the sampling sites within the city limits of Durango. Due to the Animas River’s proximity to diverse mammalian wildlife populations and homes, we hypothesized that the watershed area is likely to harbor *E. coli* of mammalian origin. The strains tested from the Animas River were essentially all thermotolerant. Thermotolerance indicates that the cultured bacteria can withstand the higher temperature of the mammalian digestive system and that the bacteria are likely of fecal origin. Further studies are needed to establish the sources of fecal-origin *E. coli*. Given the lack of ranching activity in downtown Durango, we hypothesize that sources of contamination may include wildlife, domestic pets, and/or leaking sewage conduits carrying human waste.

Testing for antibiotic resistance of the bacteria obtained from the three sites was also performed, and indicated a surprisingly high degree of multi-antibiotic resistance. Thirteen of the sixteen isolates studied showed resistance to two or three of the antibiotics tested, with penicillin being the most resisted drug. Antibiotic resistance in bacteria, particularly in infectious strains, is of growing public health concern globally [41]. Expanded testing of a larger number of bacterial isolates to better determine the prevalence of antibiotic resistant bacteria found in the Animas River would be a worthwhile project for Fort Lewis biology students to pursue in the future.

Additional drainage sites in Durango that are contributing *E. coli* to the Animas River were identified during follow-up testing in October 2015. Testing also confirmed the persistent presence of *eae*-positive strains of *E. coli*. With five of 77 isolates being positive for *eae* during October 2015 testing, and seven of 72 isolates being positive during earlier testing, a total of 12 out of 149, or 8% of all isolates tested were *eae*-positive during 2014–2015. This is noteworthy, given the role that this virulence factor plays in waterborne, diarrheal disease-related *E. coli* infections. Even if these bacteria do not possess the complete range of virulence genes needed to be able to cause disease in humans, the presence of isolates containing the *eae* virulence gene and/or expressing antibiotic resistance, indicates that the local environment harbors a subset of bacteria providing a reservoir of genes that could contribute to the evolution of fully pathogenic strains [27,42,43,44].

Biology students at Fort Lewis College have continued to isolate Escherichia coli bacteria from the Animas River drainage since initial testing in 2014. The most recent testing of these bacteria during the 2018–2019 academic school year indicates ongoing presence of *eae*-positive bacteria in the Animas River.

Microbial source tracking [45] would be an important step in determining the origin of these pathogen-related bacteria and mitigating any potential problems that may be causing spread of the bacteria, such as septic system malfunctions, or improperly managed waste. While source tracking was not performed in the present study, a geographic correlation was noted in the antibiotic resistance test results. Unique resistance patterns were observed from the Junction Creek and Skate Park testing sites. The Animas River site at Santa Rita Park site, downstream of these two tributaries, showed antibiotic resistance profiles that appear to be a combination of the tributaries’ profiles. This raises suspicions that the thermotolerant, antibiotic resistant *E. coli* found at the Junction Creek and Skate Park sites originated in unique host conditions, and that these two tributaries are the source of the antibiotic resistance phenotypes identified in bacteria at the downstream Santa Rita Park Animas River site.

In the present study, no analyses of meteorological trending or correlation were performed. It is likely that *E. coli* levels would be elevated following heavy precipitation events or snowmelt due to fecal matter lying in the surrounding hills that would drain into the Animas River. Further study is needed to verify this possible trend and establish a correlation between *E. coli* levels and seasonal weather events. While causation was not established, the elevated bacterial levels in January and February support this hypothesis. Fortunately, this is also during a time period when relatively few people would be utilizing a recreational waterway.

The drainage that exits near the Skate Park lies below the old city dump. It is unknown if there are other pollutants or contaminants in this water (or any history of water problems from the dumpsite). Forest Service workers in the area have noted that a transient population live in tents upslope from the Skate Park test site (Hamner, personal communication) and it is unknown whether that population is taking proper sanitation measures. Further study would be necessary to determine if the *E. coli* from that site are of human or animal origin.

## 5. Conclusions

The lab exercises described herein are based on previous research conducted by the authors Ford and Hamner [27] and were adapted for use in undergraduate biology and microbiology classes taught by Hamner at the University of Montana Western, Dillon, Montana in 2013 and 2016, and at Fort Lewis College, Durango, Colorado, in 2014–2015. The exercises were introduced to biology students in college communities having close connections to nearby rivers for which water quality is an issue of local importance. Many of the students of the University of Montana Western are from rural ranching families for whom water supplies are crucial. Dillon, Montana is itself in the heart of cattle country and is also home to “blue ribbon” trout streams to which many students are drawn for fly fishing and recreation. Likewise, many students of Fort Lewis College seek out rafting and fishing opportunities on the Animas River for recreation. Accordingly, many students at both schools approached the water quality studies described herein with great interest and enthusiasm. Students at Fort Lewis College were also mentored in practicing “citizen science,” presenting their findings on local water quality and discovery of *eae*-positive *E. coli* to a meeting hosted by the Animas Watershed Partnership and attended by local public health officials. Inclusion of actual research into science education programs in schools and universities that addresses topics of local concern can enhance student interest in science and provide valuable information to local communities and government officials, to help safeguard public health of local communities and protect natural resources.

## Figures and Tables

**Figure 1 ijerph-17-00195-f001:**
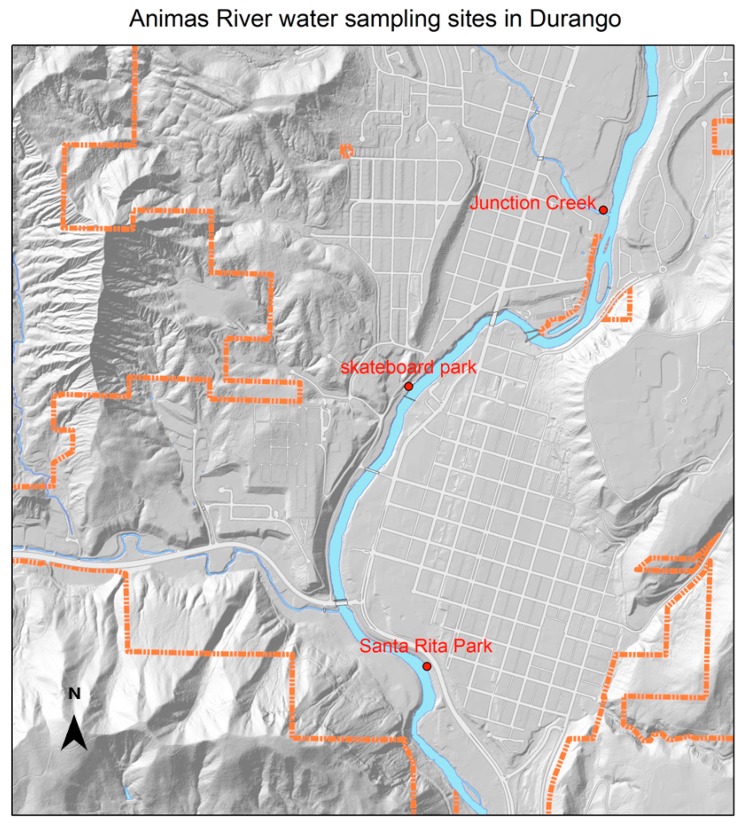
Water sampling sites on the Animas River, studied August 2014–February 2015. Santa Rita Park is located downstream of the skateboard park and Junction Creek. See Table 1 for geographical coordinates of the sampling sites. The map was created using Durango Maps (gis/durangogov.org).

**Figure 2 ijerph-17-00195-f002:**
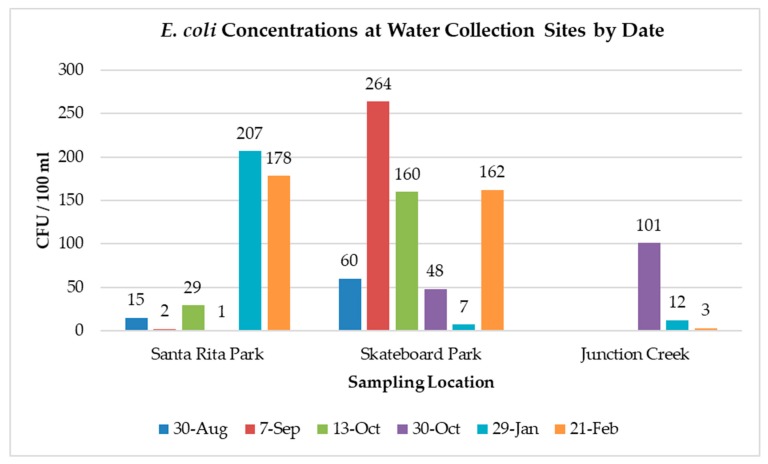
*E. coli* concentrations for the Animas River watershed, measured August 2014–February 2015. Values given are the means of triplicate counts.

**Table 1 ijerph-17-00195-t001:** Animas River sampling sites studied during 2014–2019. Geographic coordinates were captured at sampling sites using Google Earth software installed on a smartphone.

Site Name and Brief Description	Geographic Coordinates	Sampling Dates
Animas River at Santa Rita Park, “south” in Durango	37°15′35.18″ N 107°52′41.32″ W	Multiple dates during 2014–2019
Skate Park drain feeding into Animas River, Durango	37°16′39.70″ N 107°52′57.85″ W	Multiple dates during 2014–2019
Junction Creek terminus near confluence with Animas River, Durango	37°17′7.82″ N 107°52′20.17″ W	Multiple dates during 2014–2019
“West” drain feeding into Animas River, Durango	37°17′13.13″ N 107°52′18.61″ W	October 2015 only
“East” drain feeding into Animas River, Durango	37°17′18.88″ N 107°52′13.29″ W	October 2015 only
Animas River “middle” site in Durango	37°16′41.39″ N 107°52′53.87″ W	October 2015 only
Animas River “north” in Durango	37°17′8.15″ N 107°52′19.28″ W	October 2015 only
Animas River, Baker’s Bridge, upstream and north of Durango	37°27′30.08″ N 107°47′58.90″ W	October 2015 only
Junction Creek trailhead, upstream of Durango city limits	37°19′51.89″ N 107°54′11.90″ W	October 2015 only
Lightner Creek terminus near confluence with Animas River	37°16′05.10″ N 107°53′09.80″ W	2018–2019

**Table 2 ijerph-17-00195-t002:** Heat map indicating antibiotic resistance profiles of Animas River watershed bacterial isolates tested during February–March 2015 (red indicates antibiotic resistance; green indicates antibiotic susceptibility; yellow indicates some resistant colonies).

	Animas River Santa Rita Park Isolates					Skate Park Drain Isolates						Junction Creek Isolates						
	1	2	3	4		1	2	3	4	5		1	2	3	4	5	6	7
Ampicillin																		
Ciprofloxacin																		
Penicillin																		
Streptomycin																		
Tetracycline																		
Vancomycin																		

**Table 3 ijerph-17-00195-t003:** Animas River watershed testing conducted during October 2015: *E. coli* concentrations and polymerase chain reaction (PCR) test results for *eae* gene.

Site Name and Brief Description	*E. coli* Concentrations for Samples Collected on 16 October 2015 (Mean of Triplicate Counts)	Number of Isolates Positive for *eae* out of Total Isolates Tested (Sampling Conducted 30 October 2015)
Skate Park drain feeding into Animas River, Durango	91 CFU per 100 mL	2 of 19
“West” drain feeding into Animas River, Durango	149 CFU per 100 mL	0 of 16
“East” drain feeding into Animas River, Durango	Too numerous to count (about 630 colonies per 100 mL)	3 of 16
Animas River at Santa Rita Park, “south” in Durango	84 CFU per 100 mL	0 of 12
Animas River “middle” site in Durango	18 CFU per 100 mL	0 of 2
Animas River “north” in Durango	18 CFU per 100 mL	Not tested
Animas River, Baker’s Bridge, upstream and north of Durango	1 CFU per 100 mL	Not tested
Junction Creek trailhead, upstream of Durango city limits	7 CFU per 100 mL	0 of 2
Junction Creek terminus near confluence with Animas River, Durango	Too numerous to count (about 280 colonies per 100 mL)	0 of 10

**Table 4 ijerph-17-00195-t004:** Animas River watershed testing conducted during 2018–2019: PCR test results for *eae* gene.

Site Name and Brief Description, and Dates of Testing	*E. coli* Concentrations for (Mean of Triplicate Counts)	Number of Isolates Positive for *eae* out of Total Isolates Tested
Junction Creek near confluence with Animas River, Durango (October 2018)	8 CFU per 100 mL	0 of 9
Skate Park drain feeding into Animas River, Durango (October 2018)	163 CFU per 100 mL	2 of 15
Lightner Creek near confluence with Animus River, Durango (October 2018)	83 CFU per 100 mL	1 of 10
Animas River at Santa Rita Park, “south” in Durango (October 2018)	226 CFU per 100 mL	9 of 12
Junction Creek near confluence with Animas River, Durango (March 2019)	298 CFU per 100 mL	0 of 4
Skate Park drain feeding into Animas River, Durango (March 2019)	83 CFU per 100 mL	0 of 18
Lightner Creek near confluence with Animus River, Durango (March 2019)	108 CFU per 100 mL	3 of 12
Animas River at Santa Rita Park, “south” in Durango (March 2019)	69 CFU per 100 mL	2 of 11

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
