# Peer review of "Unexpected Prevalence of eae-Positive Escherichia coli in the Animas River, Durango, Colorado"

_ijerph, 2019, doi:10.3390/ijerph17010195_

Round 1
Reviewer 1 Report
Thanks for sending this through.
This partially addresses some of my original concerns. However, one comment I had made in my initial review did not get addressed:
"Whereas, two hypotheses are presented in the Introduction, the research methodology adopted is not adequate to address these (e.g. non-representative sampling, limited land-use data presented, etc.)."
Furthermore, I noticed that there is no data on column 2 of Table 4.
With this, I'd be happy to alter my decision to minor review.
Author Response
20 December, 2019
To: IJERPH Editorial Office
From: Steve Hamner and Timothy E. Ford
RE: cover letter describing responses to Reviewer #1 comments
We sincerely thank Reviewer #1 for the feedback for improving the manuscript. We very much appreciate the time and effort spent.
Our response and changes to the manuscript are described below.
Response to comments from Reviewer 1:
"Whereas, two hypotheses are presented in the Introduction, the research methodology adopted is not adequate to address these (e.g. non-representative sampling, limited land-use data presented, etc.)."
Given this concern, we have deleted the last two sentences of the last paragraph in the introduction where the two hypotheses in question were stated. In place of a hypothesis statement, we note that three goals were included in this same paragraph during the first round of response to reviewers. We believe this is an appropriate response, given that the student research was indeed guided by these goals. Please see lines 120-128. The paragraph now simply reads:
There is no noted history of pathogenic E. coli outbreaks related to bacterial contamination of the Animas River, despite the documented presence of fecal bacteria of human origin in the river. Given the reported incidents of human sewage contamination of the river, the present study was initiated with three goals in mind: to begin a targeted study of the river for presence of E. colibacteria that might pose a public health risk; to introduce biology students at Fort Lewis College to participate in an ongoing program of “citizen science” and involvement in environmental monitoring; and to share student research on water quality with local public health officials and community partners.
Furthermore, I noticed that there is no data on column 2 of Table 4.
Thank you for catching this error. Data is now included.
Reviewer 2 Report
Reviewers comments/edits have been addressed.
Author Response
Thank you for your time and effort in helping to improve the manuscript.
This manuscript is a resubmission of an earlier submission. The following is a list of the peer review reports and author responses from that submission.
Round 1
Reviewer 1 Report
The paper provides a very nice example for exposing college students to field work, with a public health-relevant case study. The strength of the paper is the detailed description of the student's involvement. The study is interesting and generalizable only for the educational component. The paper would be interesting to teachers of microbiology classes at the high school, community college, or university level.
The authors used the uid gene to indicate the presence of E. coli. However, some but not all E. coli are uid positive. EHEC do not carry the uid gene. This point needs to be clarified before the paper can be published.
There are minor word choice and grammatical improvements that can be made throughout the paper. See comments in the attached pdf.

Reviewer 2 Report
This manuscript presents results from an environmental microbiology course activity that was based on microbiological water quality monitoring. Whereas the techniques employed were adequately described with appropriate methods, some improvement on the sampling strategy could be made (i.e. a bit confusing to follow with regard to dates of visits). The text is well written in general (a bit prolix at times). This said, the manuscript contains one major shortcoming in that a clear objective or research question is not presented. This leaves it without a real purpose and also renders the conclusion pointless as there is nothing anchoring the study. Whereas, two hypotheses are presented in the Introduction, the research methodology adopted is not adequate to address these (e.g. non-representative sampling, limited land-use data presented, etc.). Instead, in its current format, the submitted manuscript reads like it would be more appropriate as contribution to a venue dedicated to instructional methods and pedagogies describing a learning activity related to environmental microbiology.